# Luminal and Mucosal Microbiota of the Cecum and Large Colon of Healthy and Diarrheic Horses

**DOI:** 10.3390/ani10081403

**Published:** 2020-08-12

**Authors:** Luis G. Arroyo, Laura Rossi, Bruna P Santos, Diego E Gomez, Michael G Surette, Marcio C Costa

**Affiliations:** 1Departments of Clinical Studies, Ontario Veterinary College, University of Guelph, University of Guelph, Guelph, ON N1G 2W1, Canada; bparapinski@gmail.com (B.P.S.); dgomezni@uoguelph.ca (D.E.G.); 2Department of Medicine, Farncombe Family Digestive Health Research Institute, McMaster University, Hamilton, ON L8S 4L8, Canada; rossil@mcmaster.ca (L.R.); surette@mcmaster.ca (M.G.S.); 3Department of Biomedical Sciences, Faculté de médecine vétérinaire, Université de Montréal, Saint-Hyacinthe, QC J2S 6Z7, Canada; marcio.costa@umontreal.ca

**Keywords:** intestinal microbiota, equine colitis, diarrhea

## Abstract

**Simple Summary:**

Acute diarrhea (colitis) is a major problem in adult horses and the role of the intestinal bacteria (microbiota) is still poorly understood in this species. The aim of this study was to compare the mucosal and luminal content microbiota of the cecum and colon of healthy and diarrheic horses. We concluded that microbial dysbiosis (changes in the normal microbiota composition) occurs in horses with colitis at different levels of the intestinal tract and microbiota composition is different between the mucosa and luminal content of diarrheic horses. Changes in species associated with dysbiosis could be used in the future for disease diagnosis, prognosis and treatment of equine colitis.

**Abstract:**

The aim of this study was to compare the mucosal and luminal content microbiota of the cecum and colon of healthy and diarrheic horses. Marked differences in the richness and in the community composition between the mucosal and luminal microbiota of the cecum and large colon of horses with colitis were observed. Microbial dysbiosis occurs in horses with colitis at different levels of the intestinal tract, and microbiota composition is different between the mucosa and luminal content of diarrheic horses. The changes in some key taxa associated with dysbiosis in the equine intestinal microbiota, such as *Escherichia*, *Fusobacterium* and *Lactobacillus*, deserve further inquiry in order to determine their utility for disease diagnosis and treatment.

## 1. Introduction

In adult horses, the acute inflammatory process of the cecum and colon, referred to as acute colitis or typhlocolitis, can result in profuse watery diarrhea, which is the hallmark clinical sign of the disease [1,2]. Horses with acute diarrhea are commonly referred to equine hospitals because they require intensive treatment [1,3,4]. Some infectious agents known to cause diarrhea include *Salmonella enterica*, *Clostridium difficile*, *Clostridium perfringens*, *Lawsonia intracellularis* (weanlings), *Neorickettsia risticii* (Potomac horse fever), coronavirus and small strongyles [1,3]. Diarrhea can also result from non-infectious causes, such as antibiotic associated diarrhea, sand impaction and toxicities (phenylbutazone or flunixin meglumine) [1,3]. However, despite extensive microbiological and epidemiological investigative efforts to determine the cause of diarrhea in horses, a causal agent cannot be established in an astonishingly large proportion (>60%) of these cases [5].

The advent of next-generation sequencing (NGS) platforms has allowed for the characterization of microbial communities from complex environments, such as the equine hind intestinal tract [6,7]. Marked perturbation of the intestinal microbiota, particularly with respect to microbial diversity, has been documented in conditions such as inflammatory bowel disease, Crohn’s disease and colitis in humans and in horses with undifferentiated colitis [7,8,9]. These observations suggest that in some cases of gastro-intestinal disorders, including horses, dysbiosis could play a role in the development of disease. A few studies using NGS showed decreased diversity and significant changes in fecal microbiota composition of diarrheic horses [7,10,11]. Differences in microbial composition between cecal and large colon contents have also been reported in healthy horses [12,13]. Similarly, a marked difference between mucosal and luminal microbiota of healthy humans and patients with gastrointestinal disorders had been observed [14,15]. These differences are driven by the presence of a mucus layer, an oxygen tension gradient and a close interaction with the immune system [16].

The luminal and mucosal microbial communities appear therefore to play distinct roles in the health and/or disease of the host gut [17,18,19]. A vast number of studies investigating the role of the intestinal microbiota of human patients suffering from diarrhea have been published [20,21]. However, although the luminal and mucosal microbiota composition of various compartments of the gastrointestinal tract of healthy horses have been reported, the microbial composition of those environments of horses with colitis is unknown [13]. The aim of this study was to compare the mucosal and luminal content microbiota in the cecum and colon of healthy and diarrheic horses.

## 2. Material and Methods

### 2.1. Animals

#### Healthy Horses

Three healthy adult horses euthanized for reasons unrelated to gastrointestinal diseases (2 had chronic arthritis and one had cervical stenosis) were used for the collection of intestinal contents and mucosal tissues. Horses were all kept on pasture without receiving any supplements, antimicrobials or anthelmintics during the 6 weeks before sampling. All horses were euthanized with a pentobarbital overdose within 24 h after arrival at the research facility. Cecal and colonic contents and mucosal tissue were collected immediately after euthanasia.

### 2.2. Colitis Cases

Cecal and colonic content and mucosal tissue from 7 horses (Table 1) presented to the Large Animal Hospital of the Ontario Veterinary College, University of Guelph, for diagnostic and treatment of acute diarrhea (1 to 3 days duration) were collected during post-mortem examination immediately after euthanasia. The breed of the horses was as follows: Thoroughbred (*n* = 3), Warmblood (*n* = 1), Quarter Horse (*n* = 1), Standardbred (*n* = 1) and mixed-breed (*n* = 1). The age ranged between 1 and 21 years.

### 2.3. Samples Collection

For mucosal microbiota investigation, approximately 2 × 2 cm full-thickness intestinal wall samples were excised during post-mortem examination from the cecum (mid-body) and left ventral colon (LVC). Intestinal contents were obtained from the same sampled locations, and all samples were collected in fecal containers and placed on ice within 60 min of euthanasia. Samples were stored at −80 °C until DNA extraction was performed. Histological examination of cecum and colon segments was undertaken in all healthy horses and colitis cases.

### 2.4. DNA Extraction and Sequencing of the V3-V4 Region of the 16S rRNA Gene

The mucosal samples were rinsed with sterile saline only once or twice to remove visible ingesta. This step was performed with care, in order to prevent any disruption of the mucus layer. DNA was extracted from mucosal tissues and intestinal content samples using the QIAamp DNA stool mini kit for pathogen detection (Qiagen, Montreal, QC, Canada) as per manufacturer’s instructions.

The DNA was diluted to a final concentration of 20 ng/μL for PCR. The 16S rRNA genes were amplified targeting the V3-V4 region [22]. The V3-V4 region of the 16S rRNA gene was amplified in a PCR reaction mixture containing 25 μL of Kapa 2G Fast Hot Start Ready Mix 2×, 1.3 μL of MgCl_2_ (50 mM) (Invitrogen, Burlington, ON, Canada), 1.0 μL of BSA (2 mg/mL) (Bio-Rad, Mississauga, ON, Canada), 16.7 μL of nuclease-free water, 2 μL of DNA and 2 μL of forward (S-D-Bact-00564-a-S-15 5′-AYTGGGYDTAAAGNG-3′) and reverse (S-D-Bact-0785-b-A-18 5′-TACNVGGGTATCTAATCC-3′) primers (10 pMol/μL).

PCR products were then purified with magnetic beads and DNA quantification was measured by spectrophotometry using the NanoDrop^®^ (Roche, Mississauga, ON, Canada). The library was pooled and sequencing was at the University of Guelph’s Advanced Analysis Centre, using an Illumina MiSeq platform using a V3 kit (2 × 300 cycles).

### 2.5. Sequence Processing and Data Analysis

Bioinformatic analysis was carried using the software Mothur v.1.39.5 [23], using a previously published protocol [24]. Good quality sequences were aligned against the SILVA database, using the Ribosomal Data Project classifier [25]. Reads were clustered at the genus level (97% similarity). Alpha diversity, which refers to the number of species (richness) and how they are distributed (evenness) within each sample was calculated based on the number of genera, Chao index (which estimates the true number of genera), Simpson and Shannon indices (which are diversity indices). Alpha diversity indices were graphically represented as strip charts generated with R!. Beta diversity (comparison of community similarity) was calculated using the Jaccard index to compare communities’ composition (unweighted, considering each genus present in those communities) and the Yue and Clayton index to compare communities’ structure (weighted, considering each genus and their abundances).

Analysis of molecular variance (AMOVA) was used to test significant differences between sites (i.e., colon vs. cecum or content vs. mucosa) and status (i.e., colitis vs. healthy). Statistical differences in the relative abundances between sampling sites and disease status were investigated using the linear discriminant analysis effect size (LEfSe) [26], which applies the Kruskal–Wallis sum-rank non-parametric test to detect differences between groups, then applying an unpaired Wilcoxon rank-sum test. The Linear Discriminant Analysis (LDA) can be set by the user to estimate the desired effect size; normally, an LDA greater than 2 is considered “biologically meaningful” or significant. In theory, the greater the LDA value, the more important that taxa would be in the analysis.

## 3. Results

### 3.1. Analysis of 16S rRNA Gene Sequencing

A total of 4,534,690 good quality reads were retained for final analysis after all bioinformatics filters were applied. Based on the sample with the lowest number of reads, a subsample of 29,373 reads was used to decrease non-uniform sample size bias during alpha diversity analysis. Average coverage after subsampling was 99.91% (SD: 0.03%), indicating that the analysis was able to detect most genera present in those samples.

### 3.2. Alpha Diversity

There were no differences in any of the alpha diversity indices, comparing all samples from healthy versus diarrheic animals. Considering only cases of colitis, intestinal mucosa had significantly greater richness than intestinal content, based on the number of observed genera (*p* = 0.001, Figure 1) and on the Chao index estimator of richness (*p* = 0.002), but not in diversity, based on the Simpson (*p* = 0.915) and Shannon (*p* = 0.813) indices (Appendix A). Noteworthily, it is evident from Figure 1 that there was much higher variability in richness (number of observed genera) among colitis samples.

### 3.3. Beta Diversity

The results of statistical comparison of beta diversity analysis comparing community composition, which takes into account the different genera present in each sample, are presented in Table 2. As expected, the healthy horses had different bacterial compositions, compared to horses with diarrhea in both luminal content and mucosa of the cecum and colon. The comparison of community composition between mucosal and luminal content revealed differences in both the cecum and colon of colitis cases, but not in cecum and colonic microbiota of healthy animals. Those differences in community composition (addressed by the Jaccard index) are clearly visualized in the principal coordinate analysis (PCoA) plots from samples collected from the cecum (Figure 2A) and colon (Figure 2B).

The comparison of community composition between cecal and colonic luminal content and cecal and colonic mucosa revealed no differences in healthy or in colitis cases (Table 2). Furthermore, there were no statistical differences in the community structure (assessed using the Yue and Clayton index) in any of the comparisons (all *p* > 0.05).

### 3.4. Relative Abundance and LefSe Analysis

The relative abundances of the bacterial genera, representing more than 1% of total reads found in the cecum and colon of healthy and diarrheic horses, are presented in Figure 3.

LefSe analysis, which searches associations between each genus and the studied groups, revealed no enriched genera differentiating luminal and mucosal microbiota in either the cecum or the colon of healthy or diarrheic horses. In addition, no enriched taxon was detected when comparing the microbiota between the cecum and colon of healthy horses. Conversely, when comparing healthy versus diarrheic horses, regardless of the intestinal compartment (colon or cecum) or the sampling site (luminal or mucosal), there were 27 taxa associated with healthy horses (LDA > 3) and 24 taxa associated with horses with colitis (Appendix A). Figure 4 represents the main taxa (LDA scores > 4) significantly associated with each group. *Lactobacillus* spp. were strongly associated with colitis, as well as *Escherichia* and *Fusobacterium* spp., which were most commonly found in two different subsets of colitis samples (Appendix A).

All the sites sampled from healthy horses were determined to be histologically normal. In all horses with colitis, the histopathologic examination revealed marked inflammation of the cecum and colon. The intestinal contents taken from all horses with colitis tested negative for the following enteropathogens: *Salmonella* enterica, *Clostridium perfringens*, *Clostridioides difficile* and *Neorickettsia risticii*.

## 4. Discussion

### 4.1. Comparison between Mucosal and Luminal Content

This study demonstrated marked differences in the richness (number of different species) and in the community composition between the mucosal and luminal microbiota of the cecum and large colon of horses with colitis. This is particularly important considering the recent findings of important crosstalk between the intestinal microbiota and the host through the production of neurotransmitters, direct neural stimulation of the enteric nervous system and through interaction with the local immune system [27,28,29]. Differences between mucosal and intestinal content microbiota have been previously reported in healthy horses [13], suggesting that, like in other species, many factors can dictate which type of bacteria can attach to and colonize the mucosa. In general, data from human studies suggest that higher diversity is expected to be found in intestinal content, compared to mucosal samples [14,30]. The present study found no differences in alpha and beta diversity between mucosal and luminal microbiota in the cecum and large colon of heathy horses. These findings were unexpected; however, it is possible that the small number of biological replicas prevented the detection of those differences.

The differences in community composition (which considers which species are shared between samples), but not in community structure (which considers also at what proportion each species is present) between content and mucosal microbiota of diarrheic horses means that they were similar overall, but it differed when the rare (or low abundant) bacteria were included in the analysis. This explains why the relative abundance plots (Figure 3) are similar between the both niches, since it represents only the main taxa present in those communities (>1% abundance). It is important to highlight that those lower abundant organisms are not necessarily less important, since they are often a source of metabolites to sustain the abundant bacteria [31].

### 4.2. Comparison between Cecum and Large Colon

Differences in the microbiota between the cecum and large colon were not observed, although they were expected. This could be explained by changes associated with the disease process, such as increased peristalsis, generalized inflammation and impaired digestion and absorption, which together could alter the normal physiology of the equine hindgut and make those two distinguished compartments more similar. Each compartment of the intestinal tract has its characteristic resident microbiota, including marked differences between the cecal and large colon in healthy horses [7,13]. Furthermore, significant differences in total and individual concentrations of volatile fatty acids (VFAs) produced in the cecum and colon have been found, which is a direct reflection of the microbial communities in each compartment [32].

Noteworthy, the characterization of confined microbiota changes is important because assessment of fecal samples may not accurately reflect the changes in other compartments. In healthy horses at least, the fecal microbiota has been shown to adequately represent bacteria from the distal gut (large and small colon and rectum) [7,13,32], but this remains to be determined in diarrheic horses. Horses with colic had significantly different microbiota in samples collected from the large colon during enterotomy, compared to fecal samples collected on hospital admission [33].

### 4.3. Comparison between Healthy and Colitis Cases

Among several other taxa, *Escherichia* spp. (and unclassified Enterobacteriaceae) and *Fusobacterium* spp. were strongly associated with samples from diarrheic horses. This enrichment was expected, as Proteobacteria have been commonly associated with dysbiosis and inflammation of the gastrointestinal tract in various species [34,35,36,37]. Dysregulated innate immune responses can elicit the blooming of Proteobacteria, which promotes gut inflammation and further promotes inflammation or pathogen invasion [38,39]. Additionally, increased fusobacteria in the microbiota of diarrheic horses been reported [12,40], and the role of these taxa deserves to be investigated in equine colitis cases. Interestingly, there was also an increased abundance of Lactobacillus within diarrheic horses, which is normally associated with health and is even used as a probiotic. Higher abundances of this genus in horses with colitis have been reported, although statistical significance was not achieved in that study [23]. Noteworthy, the descriptive nature of this study does not allow inference of causation and the association of *E. coli* and lactobacilli in cases of colitis does not necessarily mean those were causing diarrhea, but rather, may be a consequence of favorable conditions such as the depletion of other commensals, acidification of the environment and acute inflammation. This study reinforces previous findings that some bacteria normally associated with health, such as members of the Lachnospiraceae family and *Fibrobacter* spp. could be candidates considered for restoring the equine microbiota (probiotics).

Limitations of this study include the small number of horses enrolled, which comprised a heterogeneous population with wide variation in age (between 1 and 21 years old) and breeds. Nevertheless, it has been shown that the microbiota of nine-month-old horses is very similar to adults [8] and that breed might not be a great factor of variability in the equine microbiota [9]. In addition, the treatment with antimicrobial drugs received by horses with colitis prior to sample collection likely induced changes in the microbiota [10]. Nevertheless, the major findings reporting differences in alpha diversity between mucosal and luminal content were found only within diarrheic horses, and therefore it should be included as a variable, because all horses were treated with those drugs. It is possible that antimicrobials could achieve higher concentrations closer to the mucosa, but this would likely result in decreased richness, rather than greater richness compared to intestinal content, as was observed in this study. In terms of diet, horses with colitis are fed only hay while in hospital; however, details of their diets prior to admission were unknown.

## 5. Conclusions

Microbial dysbiosis occurs in horses with colitis at different levels of the intestinal tract and microbiota composition is different between the mucosa and luminal content of diarrheic horses. Shifts to key taxa associated with dysbiosis in the equine intestinal microbiota, such as *Escherichia*, *Fusobacterium* and *Lactobacillus*, deserve further inquiry in order to determine their utility for disease diagnosis and treatment.

## Figures and Tables

**Figure 1 animals-10-01403-f001:**
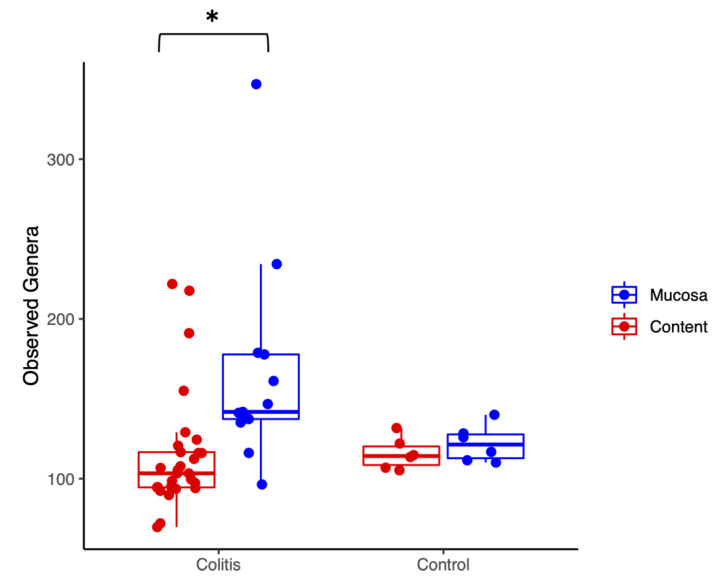
Richness indicated by the number of observed genera found in the mucosal (blue) and luminal content (red) microbiota of healthy horses and horses with colitis, demonstrating statistically higher richness in lumen compared to mucosa of diarrheic horses. * Statistical significant different between groups.

**Figure 2 animals-10-01403-f002:**
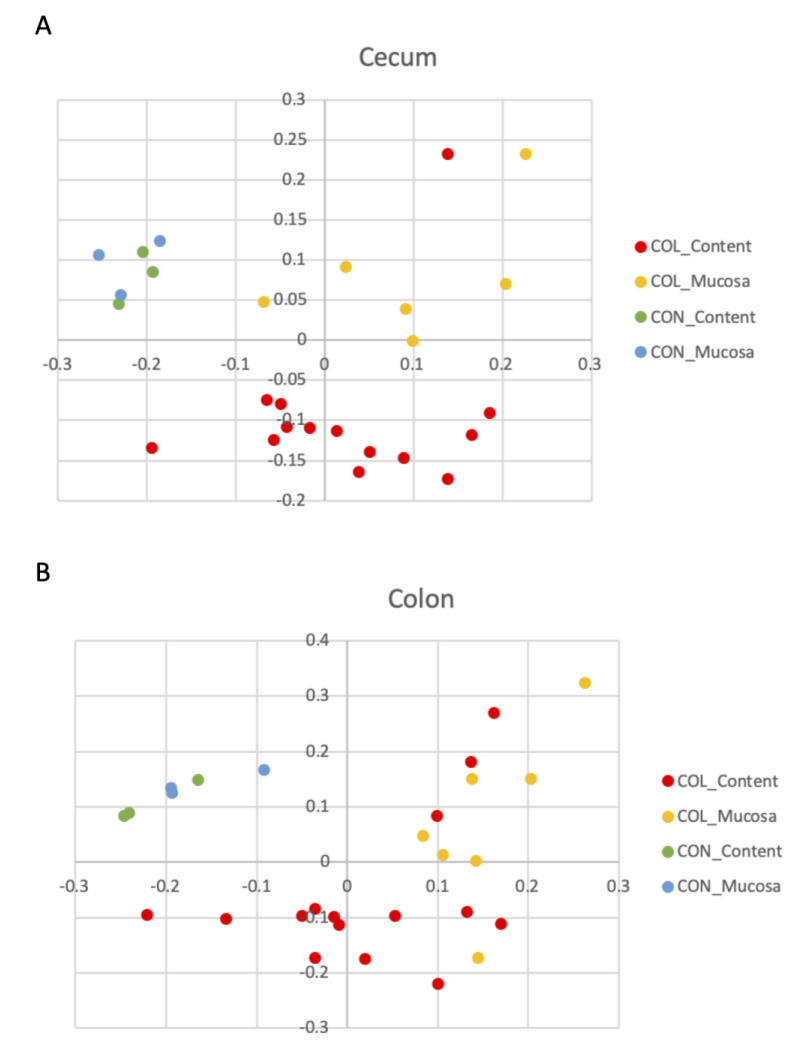
Principal coordinate analysis (PCoA) illustrating similarities in community composition of bacteria present in the lumen and content of the cecum (**A**) and colon (**B**) of healthy (CON) horses and horses with colitis (COL).

**Figure 3 animals-10-01403-f003:**
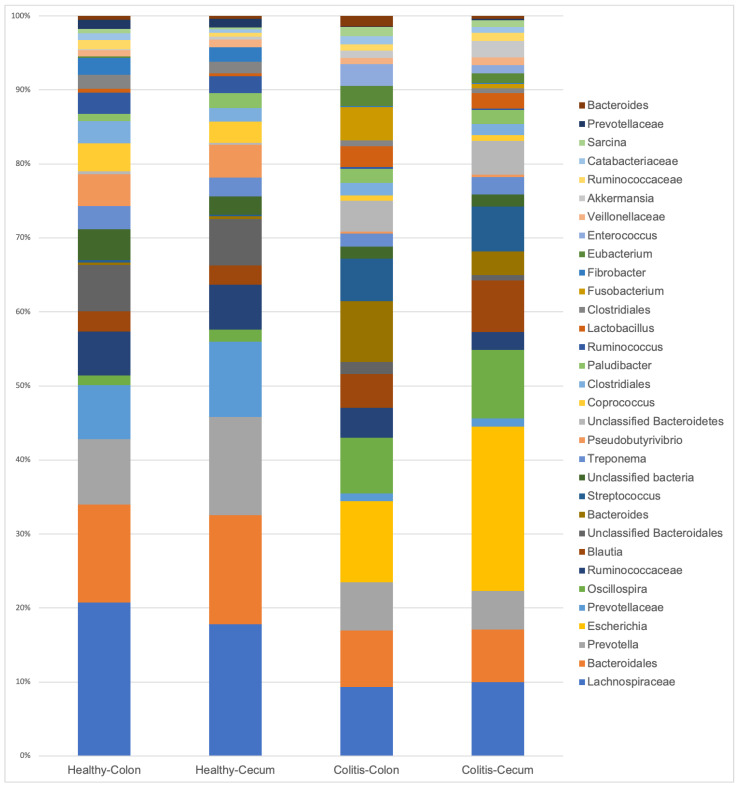
Relative abundances of the main bacterial genera (>1% of total reads) present in the cecum and colon of healthy horses and horses with colitis.

**Figure 4 animals-10-01403-f004:**
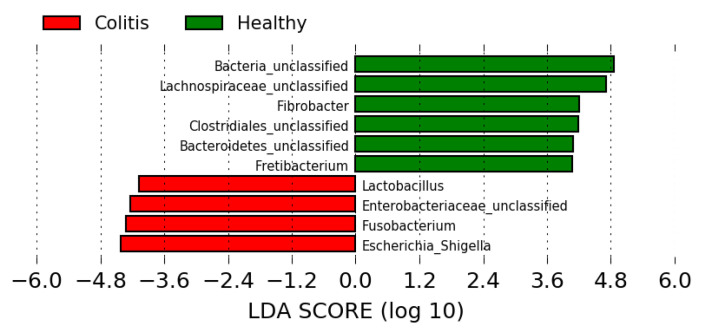
LEfSe analysis representing bacterial taxa statistically associated (LDA score > 4) with samples obtained from healthy horses (green) or horses with colitis (red).

**Table 1 animals-10-01403-t001:** Signalment, histopathological diagnosis and antimicrobials administered to colitis cases.

Bred	Age (Years)	Sex	Body Weight (Kg)	Histophatological Diagnosis	Antibiotics Administered
QH	4	MC	345	Fibrinonecrotic typhlocolitis	TMS, Pen, Gen
TB	4	Male	538	Segmental ulcerative colitis	Yes, unknown
TB	19	MC	528	Fibrinonecrotic typhlocolitis	TMS
STB	4	MC	536	Colonic edema	Pen, Gen
Belgian	21	F	702	Transmural necrosis cecum and colon	Pen, Gen, Metro
TB	1	F	380	*L. intracellularis* enteritis and necrotizing colitis	Oxytetetracycline
MB	6	F	N/A	Necrotizing and hemorrhagic ulcerative colitis	TMS, Metro, Gen, pen

QH: Quarter horse, TB: Thoroughbred, STB: Standardbred, MB: mixed-breed. MC: male castrated, F: female. TMS: Trimethoprim-Sulfametoxazol; M: Metronidazole; Gen: Gentomacin, Pen: Penicillin.

**Table 2 animals-10-01403-t002:** *p* values, degrees of freedom (df) and mean of the sum of squares (SM) obtained from the analysis of molecular variance (AMOVA) test, comparing bacterial composition present in the intestinal content and mucosa of the cecum and colon of horses with diarrhea and healthy controls.

	*p*-Value	df and (MS)	*p*-Value	df and (MS)
	**Cecum**		**Colon**	
Colitis—Healthy (content)	**0.003**	16 (0.272, 0.151)	**0.008**	17 (0.248, 0.149)
Colitis—Healthy (mucosa)	**0.010**	8 (0.301, 0.109)	**0.002**	9 (0.290, 0.136)
Content—Mucosa (Colitis)	**0.003**	19 (0.246, 0.151)	**0.034**	21 (0.242, 0.155)
Content—Mucosa (Healthy)	0.108	5 (0.137, 0.079)	0.143	5 (0.124, 0.094)
	**Healthy**		**Colitis**	
Cecum—Colon (Content)	0.673	5 (0.070, 0.084)	0.981	28 (0.097, 0.160)
Cecum—Colon (Mucosa)	0.615	5 (0.089, 0.089)	0.988	12 (0.084, 0.136)

Bold values represent *p* < 0.05.

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
