# Peer review of "Luminal and Mucosal Microbiota of the Cecum and Large Colon of Healthy and Diarrheic Horses"

_animals, 2020, doi:10.3390/ani10081403_

Round 1

Reviewer 1 Report

Well written and very interesting paper.

Here are the comments for the authors:

Line 14: I do not particularly like the term “membership”. Use appropriate microbiome vocabulary.

Line 34: There is a recent study of the equine GIT using NGS from Cohen et al from 2020. Please add reference: Álvarez-Narváez S, Berghaus LJ, Morris ERA, Willingham-Lane JM, Slovis NM, Giguère S, Cohen ND. A Common Practice of Widespread Antimicrobial Use in Horses Production Promotes Multi-drug Resistance. Sci. Reports 2020

Line 63: Did any of these horses received antibiotics as treatment for diarrhea prior euthanasia? Please explain because that could affect your results. The age range is very wide. Did you find differences in microbiome composition between younger and older horses? I know that there is literature supporting that foal’s microbiome is similar to mares’. You may want to include a brief sentence explaining this.

Line 96: You explain what B diversity is so you should explain what alpha diversity is too.

Line 113: Please add “significant” before “greater richness”. When you say “greater richness”, you mean higher numbers of observed genera. If you have more genera in one group than in the other, how the diversity is not significantly different? What about total bacterial abundance (independently from the genus), is it different between groups?

Line 116: “Noteworthy, there was much higher variability in alpha diversity indices among colitis samples”. If it is noteworthy add a plot so we can see it.

Figure 1: Please, Change the name of the plot to “alpha diversity indices” and add statistical significance. You have done a bunch of analysis. I would include a figure with the results of your Chao index. I understand that your diversity analyses were non-statistically significant but still I would include a figure with your plots in supp. Materials.

Line 122: “analysis comparing community membership”, vague term. You can use bacterial/microbes composition, microbiome composition, microbial community composition…

Line 124: “As expected, the healthy horses had different bacterial membership compared to horses with diarrhea in both, luminal content and mucosa, of the cecum and colon”.  Change membership by composition

Table 1: This table is confusing and is missing information. Please, reformat and add degrees of freedom, variance % and the statistic with P value.

Figure 2: What is the PCoA component based on? I guess Jaccard index and/or Yue and Clayton index? Please add this information to the figure. Do A and B sections correspond to: (A) community membership and (B) structure or to (A)cecum and (B) colon? Also, you named your groups as healthy and colitis but in the figure, this is not stated. Change the legend of the figure to include this information

Supplemental Figure 1: I can’t see the legend

Line 144: “LefSe analysis, which searches associations between each genus with the studied groups, revealed no enriched genera differentiating luminal and mucosal microbiota in either the cecum or the colon of healthy or diarrheic horses”. Where is this data represented? I can’t see any grouping regarding lumen or mucosa in figures 3 or 4.

Line 146: “In addition, no enriched taxon was detected when comparing the microbiota between the cecum and colon”. For both Healthy and Colitis groups? It looks like the genus Escherichia is more abundant in colitis cecum than in colitis colon while the opposite is observed for fusobacterium. This are not considered enriched?

Line 147: “Conversely, when comparing healthy versus diarrheic horses regardless the intestinal compartment (colon or cecum) or the sampling site (luminal or mucosal), there were 49 taxa associated with healthy horses (LDA > 2.5) and 43 taxa associated with horses with colitis (LDA > 2.5)”. Where is this data?  In healthy horses, enriched taxa belonged to the phyla Firmicutes (n = 26), Proteobacteria (n = 7), Bacteroidetes (n = 5), Actinobacteria (n = 1) and others (n=10). In horses with colitis, enriched taxa belonged to the phyla Firmicutes (n = 26), Proteobacteria (n = 7), Bacteroidetes (n = 4), Actinobacteria (n = 3) and others (n = 3)”. Where is this data? If it is a deeper analysis of Figure 3, why do you stay at phylum level when you have information about the genus of each taxa? Rather present a figure with the Phylum information or modify this paragraph to address the genus of the taxa. Please explain what the LDA score is

Figure 4. I would appreciate a short explanation of the meaning of the positive and negative values and colors.

Lines 168: “The intestinal contents from all horses with colitis tested negative for the following enteropathogens: Salmonella enterica, Clostridium perfringens, Clostridioides difficile and Neorickettsia risticii”. Your in silico data suggest other entero bacterias such as Escherichia and shigella to be associated with colitis. Why didn’t  you test that?

Line 172:"richness and in the community membership". Richness in terms of abundance and diversity? and community membership, you mean microbiota composition? Please rephrase

Line 181: "The present study found no differences between mucosal and luminal microbiota in the cecum and large colon of heathy horses". No differences in terms of diversity, abundance or both?

Line 184: "The differences in community membership (presence or absence of a bacterium species)" This is actually diversity.

Line 194: "Differences in the microbiota between the cecum and large colon were not observed however expected". No differences in terms of diversity, abundance or both?  Why do you think you don't see differences? please explain and add it to manuscript.

Author Response

Reviewer 1.

 Thank you for your comments and revisions.  They helped in improving our manuscript. 

Line 14: I do not particularly like the term “membership”. Use appropriate microbiome vocabulary.

AUTHORS: We were following the terminology used by the group who developed the software mothur, and have defined the term “membership” in the material and methods. We have followed your suggestion however and replaced membership by “composition”.

Line 34: There is a recent study of the equine GIT using NGS from Cohen et al from 2020. Please add reference: Álvarez-Narváez S, Berghaus LJ, Morris ERA, Willingham-Lane JM, Slovis NM, Giguère S, Cohen ND. A Common Practice of Widespread Antimicrobial Use in Horses Production Promotes Multi-drug Resistance. Sci. Reports 2020

AUTHORS: Thank you for your suggestion. The reference has been added.

Line 63: Did any of these horses received antibiotics as treatment for diarrhea prior euthanasia? Please explain because that could affect your results. The age range is very wide. Did you find differences in microbiome composition between younger and older horses? I know that there is literature supporting that foal’s microbiome is similar to mares’. You may want to include a brief sentence explaining this.

AUTHORS: Thank you for pointing that out. In fact, all cases of colitis were treated with antibiotics. A Table (table 1) describing those patients including this information has been added. We have also included a limitation statement about the use of antibiotics and population variability in the discussion.

Line 96: You explain what B diversity is so you should explain what alpha diversity is too.

AUTHORS: This information has been added.

Line 113: Please add “significant” before “greater richness”. When you say “greater richness”, you mean higher numbers of observed genera. If you have more genera in one group than in the other, how the diversity is not significantly different? What about total bacterial abundance (independently from the genus), is it different between groups?

AUTHORS: Fair point, thank you. As stated in the text, the two measures of richness used in the study were significantly greater in the mucosal samples (based on the number of observed genera (P= 0.001) and on the Chao index estimator of richness (P=0.002)”. As you know, richness and diversity are two different concepts (although both are measures of alpha-diversity). Richness refers to the number of species (either observed or estimated by the Chao index). Diversity indices are mathematical formulas that take into account the richness and the evenness (relative abundances) of each species. Thus, you can have two communities with different richness (for example one with 100 species and the other with only 10), but all of the species are equally distributed in each sample (if each species has the same abundance despite the different number of species). This is further supported by the fact that we found statistical differences in community composition (or membership) but not in structure meaning that mucosal samples might have had many more rare bacteria, but the relative abundance of the main taxa comprising both communities were similar.

Line 116: “Noteworthy, there was much higher variability in alpha diversity indices among colitis samples”. If it is noteworthy add a plot so we can see it.

AUTHORS: The variability we refer to is already represented in Figure 1. The error bars observed in cases of colitis are much wider and this is clearly seen in the figure. We have made a new picture that includes individual variation. I normally avoid Chao plots because they are redundant, since the index is based on the number of observed species, but we have added the other plots as requested (Supplementary figure 1).

Figure 1: Please, Change the name of the plot to “alpha diversity indices” and add statistical significance. You have done a bunch of analysis. I would include a figure with the results of your Chao index. I understand that your diversity analyses were non-statistically significant but still I would include a figure with your plots in supp. Materials. 

AUTHORS: We have removed the name of the plot. We have also indicated significance in the figure and added the other diversity indices as a supplementary figure, as requested.

Line 122: “analysis comparing community membership”, vague term. You can use bacterial/microbes composition, microbiome composition, microbial community composition…

AUTHORS: Changed.

Line 124: “As expected, the healthy horses had different bacterial membership compared to horses with diarrhea in both, luminal content and mucosa, of the cecum and colon”.  Change membership by composition

AUTHORS: Changed.

Table 1: This table is confusing and is missing information. Please, reformat and add degrees of freedom, variance % and the statistic with P value.

AUTHORS: The data has been added to the table.

Figure 2: What is the PCoA component based on? I guess Jaccard index and/or Yue and Clayton index? Please add this information to the figure. Do A and B sections correspond to: (A) community membership and (B) structure or to (A) cecum and (B) colon? Also, you named your groups as healthy and colitis but in the figure, this is not stated. Change the legend of the figure to include this information

Supplemental Figure 1: I can’t see the legend

AUTHORS: We are sorry for the confusion. I just noticed that the legend was wrong and it has been corrected to say that the two plots are both representing community composition from the cecum (Figure 2A) and colon (Figure 2B). We have rephrased the sentence for clarity as follows: “Those differences in community composition (addressed by the Jaccard index) are clearly visualized in the PCoA plots from samples collected from the cecum (Figure 2A) and colon (Figure 2B).”

Line 144: “LefSe analysis, which searches associations between each genus with the studied groups, revealed no enriched genera differentiating luminal and mucosal microbiota in either the cecum or the colon of healthy or diarrheic horses”. Where is this data represented? I can’t see any grouping regarding lumen or mucosa in figures 3 or 4.

AUTHORS: As stated in the text, there were NO enriched genera comparing luminal or mucosal microbiota.

Line 146: “In addition, no enriched taxon was detected when comparing the microbiota between the cecum and colon”. For both Healthy and Colitis groups?

AUTHORS: Thank you for pointing that out. We have added that this comparison was made within healthy animals.

It looks like the genus Escherichia is more abundant in colitis cecum than in colitis colon while the opposite is observed for fusobacterium. This are not considered enriched?

AUTHORS: We just noticed that Figure 4 showing results of LDA scores >3 was uploaded wrongly along with the right figure (LDA scores higher than 4). Please notice that both are showing the same data (Colitis versus Healthy) but with different cut-offs of the LDA score. The Figure with LDA scores >3 is now presented as Supplementary Figure 2.

Line 147: “Conversely, when comparing healthy versus diarrheic horses regardless the intestinal compartment (colon or cecum) or the sampling site (luminal or mucosal), there were 49 taxa associated with healthy horses (LDA > 2.5) and 43 taxa associated with horses with colitis (LDA > 2.5)”. Where is this data?  

AUTHORS: This sentence has been changed to report only LDAs >3: “Conversely, when comparing healthy versus diarrheic horses regardless the intestinal compartment (colon or cecum) or the sampling site (luminal or mucosal), there were 27 taxa associated with healthy horses (LDA > 3) and 24 taxa associated with horses with colitis (Supplementary Figure 2).”

In healthy horses, enriched taxa belonged to the phyla Firmicutes (n = 26), Proteobacteria (n = 7), Bacteroidetes (n = 5), Actinobacteria (n = 1) and others (n=10). In horses with colitis, enriched taxa belonged to the phyla Firmicutes (n = 26), Proteobacteria (n = 7), Bacteroidetes (n = 4), Actinobacteria (n = 3) and others (n = 3)”. Where is this data? If it is a deeper analysis of Figure 3, why do you stay at phylum level when you have information about the genus of each taxa? Rather present a figure with the Phylum information or modify this paragraph to address the genus of the taxa. Please explain what the LDA score is

AUTHORS: Thank you for the suggestion. The information of the different phyla was deleted and the figure with the LDA scores >3 will be kept as supplementary material, because it is more clear to read and more relevant than scores >2,5. We have also added a sentence in the material and methods to explain LDA and LEfSe.

Figure 4. I would appreciate a short explanation of the meaning of the positive and negative values and colors.

AUTHORS: The colors are explained in the left top corner legend, but we have added this information to the Figure legend as well. The negative values should be disregarded, as the absolute value of the LDA should be considered. This is how every study using this method (LEfSe) reports their data, and to our knowledge, it cannot be changed because figures are generated directly by the LEfSe website.

Lines 168: “The intestinal contents from all horses with colitis tested negative for the following enteropathogens: Salmonella enterica, Clostridium perfringens, Clostridioides difficile and Neorickettsia risticii”. Your in silico data suggest other entero bacterias such as Escherichia and shigella to be associated with colitis. Why didn’t you test that?

AUTHORS: That is a good point. Those tests were performed from samples collected at postmortem and therefore we didn’t know results at that time. The diagnosis tests used are part of a panel regularly performed in patients with colitis, which includes only the main known pathogens causing acute colitis in horses. We completely agree that more comprehensive diagnoses are necessary, but not always feasible, and E. coli is not considered a major cause of diarrhea in horses. It is important to mention that the nature of the study do not allow interpretation of cause-consequence, and therefore, E. coli and lactobacilli might be increased not because they are causing the disease, but because of an overgrowth due to dysbiosis. We have added this to the discussion. In addition, the low reliability of Illumina short reads sequencing to classify bacteria at lower taxonomic levels do not allow to distinguish between pathogenic E. coli and pathobionts, and not even distinguish E. coli from Shigella because of the their genetic similarity.

Line 172:"richness and in the community membership". Richness in terms of abundance and diversity? and community membership, you mean microbiota composition? Please rephrase

AUTHORS: We have rephased the text for clarity: “marked differences in the richness (number of different species) and in the community composition between the mucosal and luminal microbiota”.

Line 181: "The present study found no differences between mucosal and luminal microbiota in the cecum and large colon of heathy horses". No differences in terms of diversity, abundance or both?

AUTHORS: this information has been added: “no differences in alpha and beta diversity between mucosal and luminal microbiota”

Line 184: "The differences in community membership (presence or absence of a bacterium species)" This is actually diversity.

AUTHORS: I believe that this statement is correct. The Jaccard index is calculated based on a table built with all the OTUs found in the analysis and whether or not they are shared between each sample (considering only presence or absence, but not their abundances). We do agree that this sentence was very long and confusing, so it has been rephrased: “The differences in community composition (that considers which species are shared between samples), but not in community structure (that considers also at which proportion each species is present) between content and mucosal microbiota of diarrheic horses means that they were overall similar, but it differed when the rare (or low abundant) bacteria were included in the analysis.”

Line 194: "Differences in the microbiota between the cecum and large colon were not observed however expected". No differences in terms of diversity, abundance or both?  Why do you think you don't see differences? please explain and add it to manuscript.

AUTHORS: The following sentence was written in an attempt to explain this statement. It has been rephrased: “ This could be explained by changes associated with the disease process such as increased peristalsis, generalized inflammation, anomalous digestion and absorption, which together could alter the normal physiology of the equine gut and turn those two distinguished compartments more similar.”

Reviewer 2.

The authors have described an interesting study to determine changes in cecal and colon microbiota in horses with colitis. The study has straight forward methods and reports important new information about the microbiota in horses with colitis including differences in the microbiota in the mucosa of the cecum. The results should include

more clinical details including any treatments for the horses with colitis and the short-term history including any changes in diet or environment which could affect the microbiota. Characterization of the histologic changes in intestine from horses with colitis in comparison to the healthy horses should be included to rule out or discuss any

variation between horses.

Lines 58-59: Were the all the horses fed or was the diet changed before euthanasia?

AUTHORS: Thank you for your suggestion. Colitis horses are fed hay only while in hospital. Diet prior to admission was unknown. A paragraph mentioning these was added in the discussion.

Lines 63-66: Were these horses treated after admission and if so what treatments. If the treatments varied this should be included in the results and discussed.

AUTHORS: Thank you for your suggestion. A new table (table 1) has been included providing more clinical data and treatments of each patient.

Line 166-167: What were the characteristics (histological changes) of the inflammation in the horses with colitis and were these changes consistent among the horses with colitis. This should be included in the results.

AUTHORS: It was mentioned in the last paragraph of the results section that “All the sites sampled from healthy horses were determined histologically normal. In all horses with colitis the histopathologic examination revealed marked inflammation of the cecum and colon.”

Those changes were consistent and not specific of any particular pathogen.

Line 202, 216: Spelling of health.

AUTHORS: Corrected. Thank you.

Figure 2: Although intuitive please indicate what COL and CON represent.

AUTHORS: Done.

Reviewer 2 Report

+X?C?V

Author Response

Reviewer 2.

 Thank you for your comments and revisions.  They helped to improved our manuscript. 

The authors have described an interesting study to determine changes in cecal and colon microbiota in horses with colitis. The study has straight forward methods and reports important new information about the microbiota in horses with colitis including differences in the microbiota in the mucosa of the cecum. The results should include

more clinical details including any treatments for the horses with colitis and the short-term history including any changes in diet or environment which could affect the microbiota. Characterization of the histologic changes in intestine from horses with colitis in comparison to the healthy horses should be included to rule out or discuss any

variation between horses.

Lines 58-59: Were the all the horses fed or was the diet changed before euthanasia?

AUTHORS: Thank you for your suggestion. Colitis horses are fed hay only while in hospital. Diet prior to admission was unknown. A paragraph mentioning these was added in the discussion.

Lines 63-66: Were these horses treated after admission and if so what treatments. If the treatments varied this should be included in the results and discussed.

AUTHORS: Thank you for your suggestion. A new table (table 1) has been included providing more clinical data and treatments of each patient.

Line 166-167: What were the characteristics (histological changes) of the inflammation in the horses with colitis and were these changes consistent among the horses with colitis. This should be included in the results.

AUTHORS: It was mentioned in the last paragraph of the results section that “All the sites sampled from healthy horses were determined histologically normal. In all horses with colitis the histopathologic examination revealed marked inflammation of the cecum and colon.”

Those changes were consistent and not specific of any particular pathogen.

Line 202, 216: Spelling of health.

AUTHORS: Corrected. Thank you.

Figure 2: Although intuitive please indicate what COL and CON represent.

AUTHORS: Done.